# QuEChERS Extraction and Simultaneous Quantification in GC-MS/MS of Hexahydrocannabinol Epimers and Their Metabolites in Whole Blood, Urine, and Oral Fluid

**DOI:** 10.3390/molecules29143440

**Published:** 2024-07-22

**Authors:** Annagiulia Di Trana, Giorgia Sprega, Giorgi Kobidze, Omayema Taoussi, Alfredo Fabrizio Lo Faro, Giulia Bambagiotti, Eva Montanari, Maria Sofia Fede, Jeremy Carlier, Anastasio Tini, Francesco Paolo Busardò, Alessandro Di Giorgi, Simona Pichini

**Affiliations:** 1National Centre on Addiction and Doping, Italian National Institute of Health, 00161 Rome, Italy; annagiulia.ditrana@iss.it (A.D.T.); simona.pichini@iss.it (S.P.); 2Department of Biomedical Science and Public Health, Faculty of Surgery of Medicine, University “Politecnica delle Marche”, 60126 Ancona, Italy; giorgiasprega1996@gmail.com (G.S.); kobidze.giorgi@yahoo.com (G.K.); omayema.taoussi@gmail.com (O.T.); fabriziolofaro09@gmail.com (A.F.L.F.); giuliabamba@gmail.com (G.B.); evamontanari11@gmail.com (E.M.); mariasofiafede13@gmail.com (M.S.F.); jerem.carlier@gmail.com (J.C.); anastasio.tini78@gmail.com (A.T.); digiorgiale97@gmail.com (A.D.G.)

**Keywords:** hexahydrocannabinol epimers, GC-MS/MS, QuEChERS, cannabinoids, new psychoactive substances, hexahydrocannabinol metabolites

## Abstract

Recently, hexahydrocannabinol (HHC) was posed under strict control in Europe due to the increasing HHC-containing material seizures. The lack of analytical methods in clinical laboratories to detect HHC and its metabolites in biological matrices may result in related intoxication underreporting. We developed and validated a comprehensive GC-MS/MS method to quantify 9(R)-HHC, 9(S)-HHC, 9αOH-HHC, 9βOH-HHC, 8(R)OH-9(R)-HHC, 8(S)OH-9(S)HHC, 11OH-9(R)HHC, 11OH-9(S)HHC, 11nor-carboxy-9(R)-HHC, and 11nor-carboxy-9(S)-HHC in whole blood, urine, and oral fluid. A novel QuEChERS extraction protocol was optimized selecting the best extraction conditions suitable for all the three matrices. Urine and blood were incubated with β-glucuronidase at 60 °C for 2 h. QuEChERS extraction was developed assessing different ratios of Na_2_SO_4_:NaCl (4:1, 2:1, 1:1, *w*/*w*) to be added to 200 µL of any matrix added with acetonitrile. The chromatographic separation was achieved on a 7890B GC with an HP-5ms column, (30 m, 0.25 mm × 0.25 µm) in 12.50 min. The analytes were detected with a triple-quadrupole mass spectrometer in the MRM mode. The method was fully validated following OSAC guidelines. The method showed good validation parameters in all the matrices. The method was applied to ten real samples of whole blood (n = 4), urine (n = 3), and oral fluid (n = 3). 9(R)-HHC was the prevalent epimer in all the samples (9(R)/9(S) = 2.26). As reported, hydroxylated metabolites are proposed as urinary biomarkers, while carboxylated metabolites are hematic biomarkers. Furthermore, 8(R)OH-9(R)HHC was confirmed as the most abundant metabolite in all urine samples.

## 1. Introduction

Synthetic cannabinoids (SCRAs) are the most representative class of New Psychoactive Substances (NPSs), accounting for more than 250 analogues characterized up to 2023 [1]. Similarly to other NPS classes, SCRAs emerged into the drug market in the early 2000s as legal alternatives of the natural illegal cannabinoid Δ^9^-tetrahydrocannabinol (Δ^9^-THC) [2]. Indeed, SCRAs were developed from cannabimimetic analgesic compounds showing a greater binding affinity to the cannabinoid receptor CB1 than those to cannabinoid receptor CB2 [3,4]. Over the years, the illegal market uncontrollably expanded becoming a serious social and health issue due to the increasing number of related fatalities [5]. According to the European Monitoring Centre for Drugs and Drug Addiction (EMCDDA), the seizures of low-THC herbal cannabis material containing SCRAs amounted to 242 kg, 6.5 times higher than in 2020 and 1210 times higher than in 2019 [1].

Recently, the semi-synthetic cannabinoids subclass has been raising concerns due to the increased popularity of new analogs, such as Δ8-THC and hexahydrocannabinol (HHC) [6,7,8]. Although it was discovered in 1940 by Adams et al., HHC emerged on the drug market in the United States (US) in late 2021, while its first identification as a drug of abuse dates to May 2022. Following the report of HHC-containing product seizures in 20 EU Member States, the EMCDDA posed under strict control the HHC as new a NPS by March 2023 [8,9]. HHC is easily synthesized from cannabidiol (CBD), which in turn is extracted from low-THC cannabis [7].

HHC is characterized by a hexahydro cyclohexyl ring structure and it naturally occurs as two different epimers depending on carbon 9’s stereochemistry, 9(R)-HHC and 9(S)-HHC (Figure 1) [3]. Higher “cannabis” effects were observed for 9(R)-HHC, while the 9(S)- epimer did not show activity even at high doses [7].

As for all the NPSs, the HHC analytical detection in biological specimens represents a crucial step for its pharmacological profiling and a fundamental tool to confirm HHC intoxications in laboratory medicine. Furthermore, considering the different pharmacological activity of 9(R)-HHC and 9(S)-HHC, the resolution of the epimers may represent an advantage for the clinical and toxicological laboratories, allowing us to determine the epimeric composition of the administered drug while looking for the parent drugs and the metabolites as biomarkers of exposure. Presently, a few methods were developed to detect HHC epimers and their metabolites in biological matrices using various analytical techniques such as GC-MS [10], HPLC-HRMS/MS [11], HPLC-MS/MS [12], or immunological screening [13]. The GC-MS method [10] allowed the simultaneous detection of parent compounds and metabolites in urine after a solid-phase extraction (SPE). However, the low sensitivity and the high method duration represent a disadvantage for high-throughput and routine laboratories. Better sensitivity was achieved through a GC-MS/MS assay, which was insufficient to quantify all the detected metabolites. Contrastingly, an untargeted HPLC-HRMS/MS method [11] allowed the elucidation of a wider range of metabolites in urine, providing interesting insights on the HHC metabolism. The analytes were extracted from 800 µL urine through a liquid–liquid extraction (LLE) using n-butyl acetate. However, the high maintenance costs and the highly specialized personnel required make it difficult to use in routine laboratories. Two different HPLC-MS/MS methods [12] were developed to quantify parent compounds and HHC metabolites, respectively, in a small volume of urine, oral fluid, and blood. A unique LLE protocol using 3 mL hexane:ethyl acetate 9:1 (*v*/*v*) was applied to both methods. Moreover, the analytes’ chromatographic separation was obtained through a chiral stationary phase column. Good sensitivity, accuracy, and precision were achieved. Unfortunately, chiral stationary phase chromatographic columns are expensive. In this context, we aimed to develop a comprehensive method to reliably quantify HHC epimers and eight epimeric metabolites, using a time and cost-effective extraction procedure coupled with common instrumental equipment for toxicological, clinical, and emergency department laboratories.

Differently from the other methods, we applied, for the first time, a modified QuEChERS extraction protocol to simultaneously quantify 9(R)-HHC, 9(S)-HHC, 9αOH-HHC, 9βOH-HHC, 8(R)OH-9(R)-HHC, 8(S)OH-9(S)HHC, 11OH-9(R)HHC, 11OH-9(S)HHC, 11nor-carboxy-9(R)-HHC (11nor-9(R)COOH HHC), and 11nor-carboxy-9(S)-HHC (11nor-9(S)COOH HHC) in a reduced volume of whole blood, urine, and oral fluid (OF). Considering the current knowledge on 9(R)-HHC and 9(S)-HHC human metabolism [10] and similar cannabinoids [14], we included all the epimer couples of each possible metabolite that were available as certified analytical standard solutions on the market. The method was fully validated proving to be rapid and cost-effective, thanks to the reduced steps for the extraction, allowing us to process a large number of samples per analytical batch with instrumental equipment commonly available in clinical and toxicology laboratories. In addition, the limited use of organic solvent made the method environmentally friendly and safer for personnel. Finally, the method was applied to quantify the epimeric metabolites and the parent drugs in 10 real samples from HHC users, allowing us to disclose important features and differences in the metabolic profile of the 9(R)-HHC and 9(S)-HHC.

## 2. Results and Discussion

### 2.1. Method Development and Validation

The method allowed the efficient and rapid extraction and quantification of all the target analytes in all three investigated matrices in a 12.50 min chromatographic run (Figure 2).

Furthermore, we successfully separated all the epimers through a non-chiral capillary column, with good results for the isobaric hydroxylated metabolites also. The best instrumental conditions were developed by injecting separately the analytical standard solution of each target compound into the gas chromatographer, before and after the derivatization with BSTFA. To this concern, no signal was observed before the derivatization, while an acceptable signal was observed when applying the derivatization protocol routinary used for cannabinoid detection in our laboratory. Surprisingly, the 9αOH-HHC appeared differently derivatized as a mono-O-TMS derivative, showing a [M^+^] of 404 *m*/*z* (Table 3). Probably, the steric encumbrance due to the tridimensional arrangement of methyl and OH group on position 9 impeded the silanization of the site (Figure 1). To this concern, the trimethylchlorosilane percentage in the derivatization agent could be increased for further derivatization of the compound. Originally developed for pesticide detection in food, the QuEChERS extraction was demonstrated to be suitable for a variety of illicit compounds in different biological matrices, such as whole blood, urine, or breastmilk [15,16,17]. In this regard, a unique QuEChERS-based extraction was established for all the investigated biological matrices with analytical recovery results ranging between 81.7 and 110.7% in whole blood, 85 and 107% in OF, and 98.2 and 116.9% in urine. To this concern, different salt compositions were tested by extracting three QCs and evaluating the recovery rates. In particular, Na_2_SO_4_:NaCl 4:1 (*w*/*w*), Na_2_SO_4_:NaCl 1:1 (*w*/*w*), and Na_2_SO_4_:NaCl 2:1 (*w*/*w*) were the tested mixtures. While the 1:1 (*w*/*w*) mixture provided low recovery percentages (range 57–71%) for parent compounds, the 4:1 mixture (*w*/*w*) yielded unsatisfying rates for metabolites. The best performance compromise was obtained with the 2:1 (*w*/*w*) mixture, with satisfying recovery rates for all the target analytes (Table 1). Moreover, the QuEChERS purification step was tested using the primary–secondary amine (PSA) as a purification sorbent, but recovery rates did not show significant improvements. For this reason, this step was avoided, saving time and costs. The highest recovery results were observed in urine, suggesting an ion enhancement from the matrix that was never observed for blood. The QuEChERS protocol showed good versatility and allowed us to extract all ten target analytes from a small amount of matrices (three) with very different physicochemical properties. Compared to the most used extraction techniques for cannabinoids such as solid–liquid extraction, the QuEChERS extraction was very rapid, cheap, and eco-friendly since all the chemicals implied in the extraction were selected according to the green chemistry principles. Furthermore, it allowed the extraction of all analytes in one step, without the application of any buffer or acidic/basic solution to adjust the pH. Unfortunately, the derivatization step could not be avoided due to the presence of hydrophilic moieties on the analytes, which affect the volatility of the compounds. Further studies could be conducted to develop a green alternative to silanization in GC-MS/MS analysis.

Since scarce information on the HHC pharmacological profile is still available [10], the inclusion of all the commercially available HHC putative metabolites was fundamental to develop a comprehensive method suitable for routine analyses in clinical laboratories and pharmacological studies. To this concern, we studied whole blood and urine, which are considered the principal matrices in the clinical and laboratory medicine fields, and OF as one of the most promising alternative matrices [18]. To this concern, the drug transition into the OF from the blood depends on different factors such as the physicochemical characteristics of the drug, the pH of blood and OF, the fraction bound to plasma proteins, and the salivary flow rate [19]. No interferences were observed from the matrices neither from the deuterated standards after the analyses of blank and negative samples of all the matrices (Figure 3). Furthermore, carryover was not observed for any analytes.

The calibration range for each substance was set up considering the preliminary analyses of real samples obtained with an operative calibration curve set up from 1 ng mL^−1^ up to 500 ng mL^−1^ in all matrices, applying the same instrumental conditions and the same extraction protocol. Therefore, the optimal calibration range was developed to comprise the concentration range of real samples, obtaining more reliable results (Table 1). According to the OSAC guidelines' criteria [20], the method showed acceptable validation parameters for all the analytes in all the evaluated matrices within the calibration range. To this concern, the method showed good linearity with a *p*-value ranging from 0.1010 to 0.8650, and a TV value below the Fcrit value [18.5] for all the curves. Bias, within run precision and between run precision, was within the acceptable criteria for all the analytes in all the matrices. However, the method exhibited lower sensitivity (LOD = 0.8 ng mL^−1^), accuracy, and precision (bias between 12.1 and 19.5% and precision between 12.2 and 18.2%) at lower concentrations in urine. Conversely, the best results were observed for OF.

### 2.2. Real Samples Results

The method was applied to 10 anonymized real samples of urine (n = 3), blood (n = 4), or oral fluid (n = 3) collected from HHC consumers at the University Politecnica delle Marche (Table 2). Unfortunately, no information on the single cases was available avoiding a precise interpretation of the toxicological findings. However, the results confirmed a different concentration rate for the two epimers, corroborating the hypothesis of a different metabolic fate (Figure 4) [21]. In general, the 9R epimers were predominant in all the samples, suggesting that the primary drug contained a higher percentage of 9(R)-HHC, the most psychoactive epimer. Furthermore, the metabolites appeared differently distributed in the considered matrices.

### 2.3. Blood Samples

The four different blood samples showed a relatively low concentration of the parent drug, with 9(R)-HHC being the predominant epimer (P1 = 1.4 ng mL^−1^ and P6 = 1.6 ng mL^−1^), while 9(S)-HHC was detected only in patient 7 with a 9(R)/9(S) ratio of 2.26. Interestingly, the HHC epimers were not detected in one sample, which contained only 11nor-COOH-9(R)-HHC and its metabolic precursor 11OH-9(R)-HHC. Supposedly, the sample was collected later than the others, since the HHC was excreted and metabolized. To this concern, a low quantity of 9(R)-HHC was detected alone in two samples in which the carboxy metabolite was the most abundant, with the 9S epimer detected in minimum quantities only in one sample. Although the 8(R)OH-9(R)HHC appears controversial since it was not always detected, it showed the highest average concentration in blood samples, which was first observed as a minor metabolite in urine [10,12,21]. Notably, all the metabolites were detected as glucuronic acid conjugates, while they were analytically observed only after enzymatic hydrolyzation.

### 2.4. Oral Fluid

As expected, the highest concentration of the parent drug was detected in OF samples, while metabolites were not revealed. In particular, sample P4 was reanalyzed after dilution to fit the calibration curve. Considering the recent highlight on the HHC pharmacokinetic, it is likely that the psychotropic drug was taken a few hours before the sampling or it was administered at a high dosage [21]. Although information on the abused substance was not available, it is interesting to note that P2 and P3 showed a similar 9R/9S ratio (2.5), while P4 presented a doubled ratio, suggesting a different composition in the epimers of the drug. Hence, OF was confirmed as a suitable matrix to prove the HHC consumption of both epimers, using also less sensitive analytical techniques and avoiding the hydrolyzation step in sample preparation.

### 2.5. Urine

All the urine samples were unambiguously positive for HHC since both the epimers and different metabolites were quantified in all the samples. Notably, sample P9 presented a higher concentration of 9S epimers than that of 9R and the ratio 9R/9S was not consistent among the considered samples. Similarly to blood, the metabolites were detected only as glucuronides since no signal was detected in any sample when hydrolysis was not performed. While 8(R)OH-9(R)HHC and 11nor COOH-9(R)HHC were quantified in all the samples, different 9S metabolites were observed in each urine sample, suggesting a great interindividual difference in the metabolism of 9(S)-HHC. As already observed, the highest concentration was observed for 11OH-9(R)-HHC, although it was not observed in sample P8 [10,22]. To this concern, a complete further metabolization in the 11-carboxy metabolite is supposed due to rapid metabolization.

## 3. Materials and Methods

### 3.1. Reagents and Chemicals

All the solvents were of analytical grade. The analytical standards of the 9(R)-HHC, 9(S)-HHC, 9αOH-HHC, 9βOH-HHC, 8(R)OH-9(R)-HHC, 8(S)OH-9(S)HHC, 11OH-9(R)HHC, 11OH-9(S)HHC, 11nor-9(R)COOH HHC, 11nor-9(S)COOH HHC (1mg mL^−1^ methanolic solution) and the deuterated standard (Δ^9^-THCd3 and THC-COOH d3, 1mg mL^−1^ methanolic solutions) were purchased from Cayman Chemicals (Ann Arbor, MI, USA). The magnesium sulfate and the sodium chloride were purchased from Carlo Erba (Milano, Italy). The N,O-Bis(trimethylsilyl)trifluoroacetamide with trimethylchlorosilane (99:1, BSTFA) for derivatization and β-Glucuronidase from Helix pomatia (≥100,000 units mL^−1^) aqueous solution were purchased from Sigma-Aldrich. (Saint Louis, Mo, USA)

### 3.2. Human Blank Samples and Real Samples

The pooled blank human samples of blood, urine, and oral fluid, used for blank samples, calibrators, and quality control (QC) samples’ preparation, were obtained from the laboratory storehouse of blank samples, in compliance with the institutional protocol. All the matrices were analytically confirmed as blank by routine GC-MS general screening before the validation experiments. The 10 real samples of actual HHC users were kindly donated by the Department of Biomedical Science and Public Health of the University “Politecnica delle Marche”.

### 3.3. Standard Solution Preparation

Eight working standard methanolic solutions (WSTDs) with all 10 target analytes for the calibrators and QC samples (highQC, mediumQC, and lowQC) were prepared and stored at −20 °C until analyses were carried out. The upper calibrator WSTD (WSTD5), the highQC WSTD, and the WSTD for dilution integrity assessment were obtained by diluting the analytes stock solution in methanol to obtain the concentration reported in the Appendix A. The other calibrators, WSTD, were obtained for the subsequent dilution of the more concentrated WSTD. Similarly, the QC WSTD solutions were prepared for subsequent dilution of the highQC WSTD. The final concentration of all spiked samples and the relative WSTD solution are reported in Appendix A. Since the analyte deuterated standards were unavailable on the market, Δ^9^-THC_d3_ and THC-COOH_d3_ were selected as the most chemically similar molecules to the target analytes, with basic and acidic properties. To this concern, the internal standard (ISTD) solution was prepared diluting a proper volume of the methanolic standard solution of Δ^9^-THC_d3_ and of THC-COOH_d3_ up to a concentration of 2 µg mL^−1^.

### 3.4. QuEChERS Extraction Protocol

Calibrators, QCs, and real samples were prepared according to the same QuECHERS extraction protocol, depending on the matrix. In particular, 200 µL of whole blood or urine spiked with 10 µL of ISTD, and 10 µL WSTD in the case of calibrators and QC samples, was added with 100 µL acetate buffer 0.1 M at pH 5 and 40 µL of β-glucuronidase solution and was incubated at 60 °C for 2 h. The hydrolysis step was adapted from the routine protocol analysis of cannabinoids currently in use in our laboratory. Then, the hydrolysis was stopped by adding 500 µL of acetonitrile and centrifuging the samples at 4500 rpm for 5 min. The supernatant was mixed with 45.71 mg NaCl and 85.71 mg MgSO_4_ for 10 min with a rotative mixer at high speed. After, the samples were centrifuged at 4500 rpm for 5 min and the supernatant was transferred to a clean tube and dried at room temperature under gentle nitrogen flow. Finally, the derivatization was performed by adding 50 µL of BSTFA. Concerning the OF samples, the same procedure was applied, excluding the hydrolyzation step since glucuronides are not excreted in OF [19]. Dilution integrity samples were properly diluted with polled blank matrix and 200 µL was processed as reported above.

### 3.5. Instrumental Conditions

The chromatographic separation was carried out through an HP-5ms Ultra inert column (30 m, 0.25 mm × 0.25 µm, Agilent Technologies, Santa Clara, CA, USA) on a 7890B GC (Agilent Technologies, Santa Clara, CA, USA) system equipped with a multimode injector, working in a pulsed splitless mode at 270 °C. Helium was used as a carrier gas at a flow of 2.25 mL min^−1^. The column oven was initially held at 120 °C for 1 min and then increased until 270 °C with a 20 °C min^−1^. After 2 min, the temperature was ramped with a rate of 30 °C min^−1^ up to 300 °C, and maintained for 1 min. The 7000C triple quadruple mass spectrometer (Agilent Technologies, Santa Clara, CA, USA) equipped with an electron impact ionization source was set in the multiple reaction monitoring (MRM) mode, using N_2_ at a flow of 1.5 mL min^−1^ as a collision gas. A qualitative and a quantitative transition were selected for each analyte, while a single MRM transition was selected for ISTD (Table 3).

### 3.6. Method Validation

The method was fully validated for bias, linearity, carryover, selectivity, sensitivity (limit of detection, LOD, and Limit of Quantification, LOQ), precision, dilution integrity, and stability in the three matrices according to a five-day protocol, following the most recent recommendations for method validation in the Forensic Toxicology of the Organization of Scientific Area Committees (OSACs) [20]. Briefly, selectivity was evaluated by assessing the possible interferences from the matrix by analyzing a pooled blank matrix and monitoring the absence of any signal of the analytes, whereas the analysis of the pooled blank matrix fortified with the ISTD allowed us to assess the presence of interfering signals from the deuterated standards. Bias was measured by assessing the three QC samples over five different runs, considering the maximum acceptable value as ±20%. The within-run precision was calculated as the coefficient of variation (CV%) of the concentration of each QC sample injected in triplicates for each run, while the between-run precision was calculated on the CV% of the three QC samples injected over the five runs. The maximum accepted value was 20%. The method linearity was assessed by injecting the calibrators samples in triplicates for five consecutive runs and performing the Mandel test for linearity with a confidence level of 95% [23]. Carryover was assessed by injecting drug-free samples of each matrix after the highest point of the calibration curve in each analytical batch. The LOQ was experimentally determined and set as the lowest non-zero calibrator of each calibration curve, while the LOD was experimentally determined by analyzing spiked samples with decreasing concentrations of the analyte and thereafter calculating the signal-to-noise ratio, with an acceptance rate of 3.3 and acceptable predefined detection criteria. Dilution integrity was determined by assessing the precision and accuracy of spiked samples with nominative concentrations 2, 5, and 10 times over the calibration curve, after proper dilution. Analytical recovery was determined at the QC points, following the experimental design proposed by Matuszewski et al. [24]. Specifically, set 1 was composed of 5 replicates of standard analytes at each QC concentration; sets 2 and 3 were composed of 5 blank samples fortified after and before extraction, respectively, at the same concentration of set 1. Then, for each analyte and QC point, analytical recovery was calculated by dividing mean peak areas of set 3 by set 2. Acceptable criteria were ±20% of the target concentration.

## 4. Conclusions

The recent emergence of the semi-synthetic cannabinoid HHC posed a new challenge for the analytical laboratory devoted to toxicological and clinical analysis, since scarce and controversial information on pharmacokinetics is available so far. Furthermore, the natural composition of the drug of abuse in two different epimers, 9(R)-HHC and 9(S)-HHC, is a challenging aspect for routine laboratories, which should discriminate the most potent epimer 9(R)-HHC from 9(S)-HHC presenting a different metabolic profile. To this end, we successfully developed a fast and cost-effective analytical method in GC-MS/MS to sensitively quantify the parent drug epimers and eight different metabolites in a single chromatographic run, applying the same extraction protocol. In particular, the QuECHERS extraction proved to be suitable for urine, blood, and OF analysis with the advantage of reducing the organic solvent usage, resulting in more cost-effective and eco-friendly extraction than the usual LLE or the SPE. Finally, the method was applied to analyze 10 different real samples from HHC consumers confirming the different metabolization and distribution of HHC epimers in the considered biological matrices. Whereas, the parent drug epimers were detected in OF, urine, and blood samples and were presented only in phase II metabolites as glucuronides. To this concern, 8(R)OH-9(R)HHC and 11nor COOH-9(R)HHC appeared as the best 9(R)-HHC biomarkers in urine and blood, while glucuronides 9(S)-HHC was the best biomarker in urine for 9(S)-HHC intake.

## Figures and Tables

**Figure 1 molecules-29-03440-f001:**
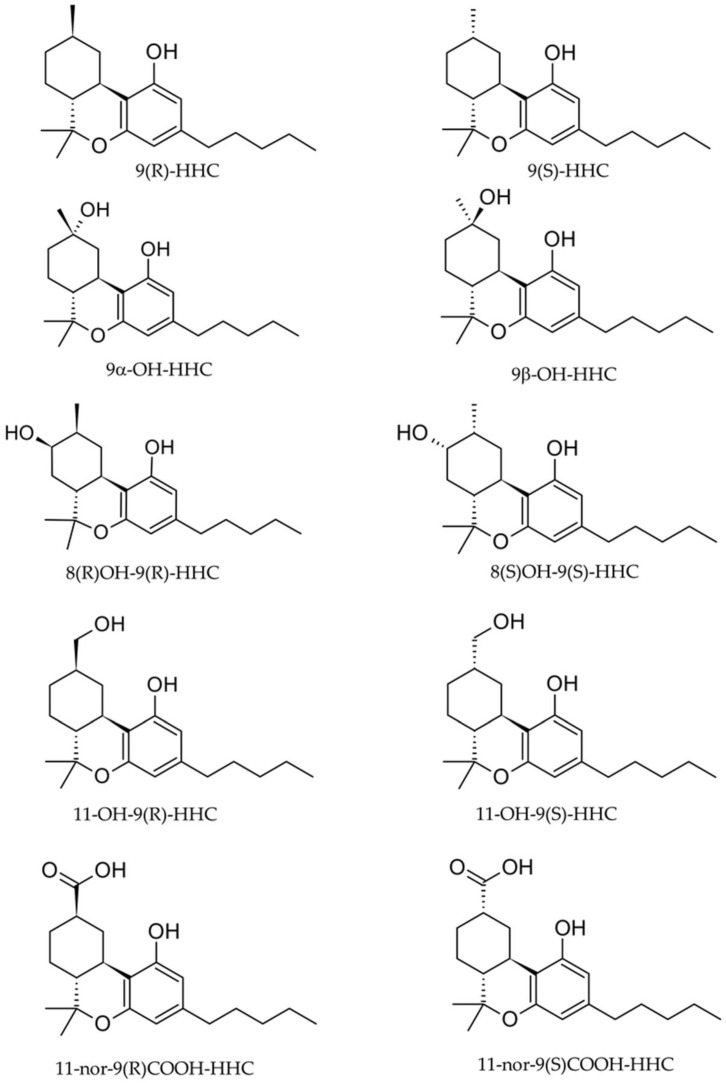
Molecular structures of the ten target analytes.

**Figure 2 molecules-29-03440-f002:**
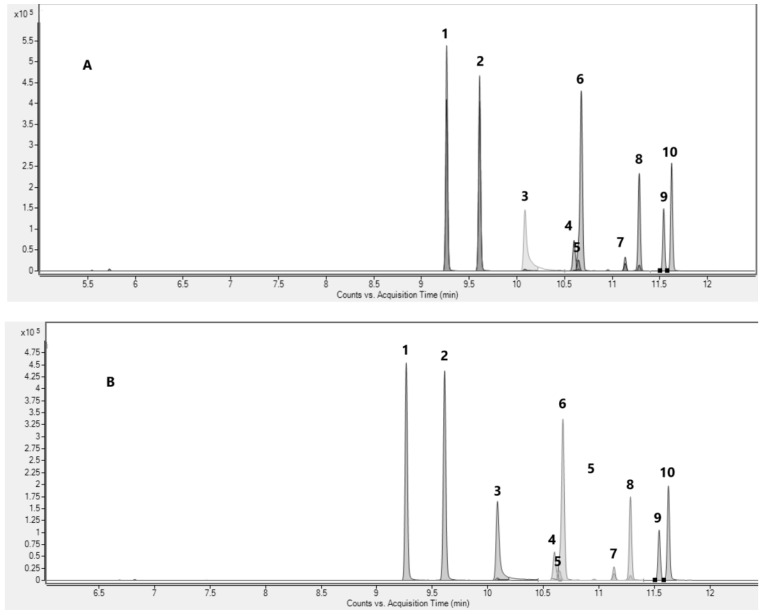
Representative extracted MRM chromatogram of the target analytes in spiked oral fluid (**A**), urine (**B**), and whole blood (**C**). Legend: (1) 9(R)-HHC; (2) 9(S)-HHC; (3) 9αOH-HHC; (4) 8(R)OH-9(R)-HHC; (5) 9βOH-HHC; (6) 8(S)OH-9(S)HHC; (7) 11OH-9(R)HHC; (8) 11OH-9(S)HHC; (9) 11nor-carboxy-9(R)-HHC; and (10) 11nor-carboxy-9(S)-HHC.

**Figure 3 molecules-29-03440-f003:**
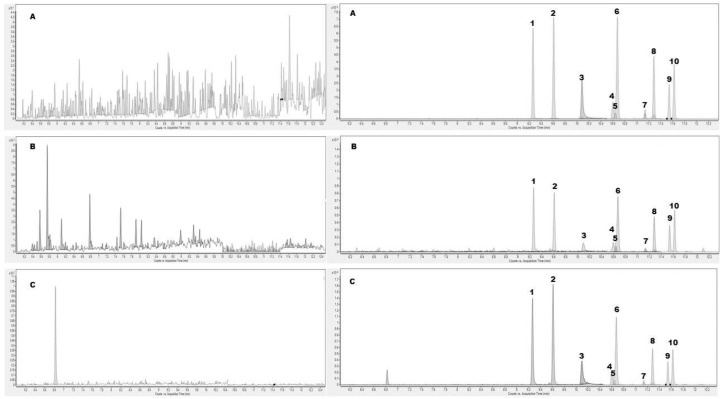
Representative baseline-scaled extracted ion chromatogram of the pooled blank matrix (on the **left**) of oral fluid (**A**), urine (**B**), and whole blood (**C**) compared to the extracted ion chromatogram of the spiked matrix at the Limit of Quantification for all the target analytes (on the **right**). Legend: (1) 9(R)-HHC; (2) 9(S)-HHC; (3) 9αOH-HHC; (4) 8(R)OH-9(R)-HHC; (5) 9βOH-HHC; (6) 8(S)OH-9(S)HHC; (7) 11OH-9(R)HHC; (8) 11OH-9(S)HHC; (9) 11nor-carboxy-9(R)-HHC; and (10) 11nor-carboxy-9(S)-HHC.

**Figure 4 molecules-29-03440-f004:**
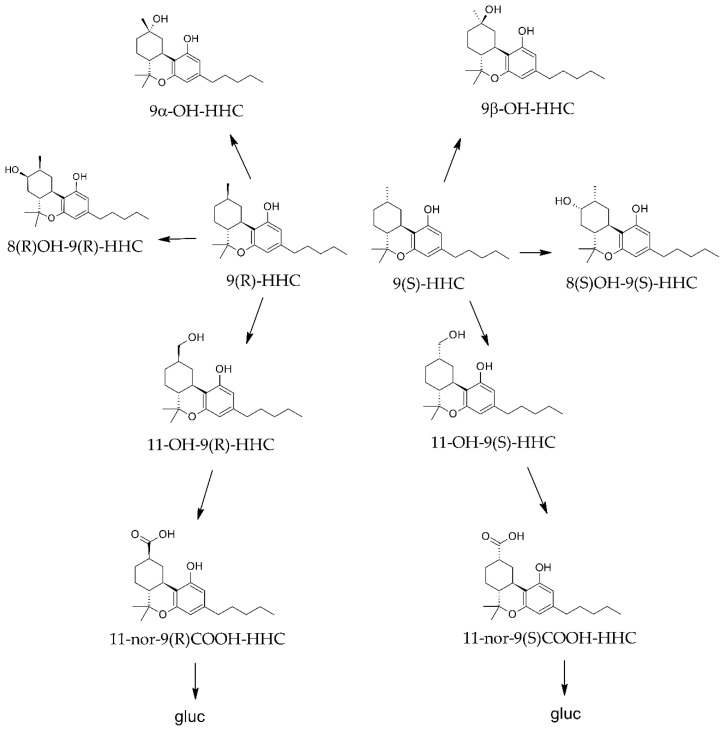
Putative metabolic pathways of HHC epimers.

**Table 1 molecules-29-03440-t001:** Validation parameters assessed according to OSAC guidelines for the ten target analytes in whole blood, urine, and oral fluid were extracted with a QuEChERS-based extraction protocol. The determination coefficient (r^2^), TV, and *p* refer to the linearity of the method within the calibration range, as reported in the section results. The accuracy and precision were assessed at three concentrations. Acceptance criteria OSAC guidelines: bias, ±20%; within-run precision, ±20%; intra-assay precision, ±20%; and recovery, ±20% of the target concentration.

Analyte	ISTD	Matrix	Calibration Range	r^2^	TV	*p*	LOD	LOQ	Bias (±%err)			Within Run Precision (CV%)			Intra-Assay Precision (CV%)			Recovery
							(ng mL^−1^)	(ng mL^−1^)										
			(ng mL^−1^)						lQC	mQC	hQC	lQC	mQC	hQC	lQC	mQC	hQC	Average (CV%)
9(R)-HHC	Δ^9^-THCd_3_	Blood	1–30	0.999	0.037	0.865	0.5	1	8.1	2.1	4.5	12.1	3.8	6.6	14.6	9.9	12.1	82.1
		OF	1–150	0.997	5.954	0.135	0.2	1	11.9	3.7	5.3	10.1	5.6	3.4	13.2	9.8	7.8	101.6
		Urine	1–150	0.996	6.971	0.118	0.8	1	15.1	6.4	12.3	14.3	7.3	11	9.7	9.8	6.9	114.3
9(S)-HHC	Δ^9^-THCd_3_	Blood	1–30	0.988	5.599	0.11	0.5	1	7.1	5.3	9.1	10.8	2.3	1.2	9.5	7.5	12.1	90.8
		OF	1–150	1.000	0.373	0.603	0.2	1	12	4.8	6.1	9.4	4.1	4	12.3	6.1	14.6	99.2
		Urine	1–30	0.999	2.728	0.240	0.8	1	15.5	7.8	13.2	13.2	9.1	9.9	17.2	6.9	7.5	113.4
9αOH-HHC	Δ^9^-THCd_3_	Blood	1–30	0.998	0.038	0.864	0.5	1	17.6	7.8	16.1	12.4	2.2	2.9	9.6	5.4	10.1	84.2
		OF	1–30	1.000	0.75	0.478	0.8	1	16.1	7.7	18.1	15.1	3.9	3.9	11.4	4.3	9.8	85
		Urine	1–150	0.985	2.23	0.468	0.8	1	19.5	10.2	15.6	18.2	4	8.3	10.3	5.9	9.8	98.2
9βOH-HHC	Δ^9^-THCd_3_	Blood	1–30	0.998	5.317	0.103	0.5	1	11	6.7	8.9	9.8	6.6	9.1	12	7.9	10.1	97.8
		OF	1–50	0.999	1.682	0.324	0.5	1	10.9	4	7.7	9.3	5	5.3	9.8	6.6	7.3	97.7
		Urine	1–30	0.986	0.23	0.677	0.8	1	16.7	8.1	9.9	8.7	7	7.7	10	9.1	12.1	116.9
8(S)OH-9(S)HHC	Δ^9^-THCd_3_	Blood	1–30	0.985	3.517	0.199	0.5	1	9	2.1	6.4	11	4.6	6.8	12.6	7	10	110
		OF	1–30	0.999	6.174	0.109	0.2	1	5.4	2.3	11.1	7.9	3.1	2.1	15.1	6.5	7.9	88.9
		Urine	1–150	0.997	2.311	0.268	0.8	1	13.6	6.9	13.7	12.2	6.7	12	14.3	7.4	8.3	112.2
8(R)OH-9(R)HHC	Δ^9^-THCd_3_	Blood	1–50	0.997	5.146	0.108	0.5	1	17.3	8.9	5.1	8.9	8.3	7.9	8.5	9.3	7.5	88.9
		OF	1–30	0.997	5.119	0.154	0.5	1	15.8	6	7.8	10	2.1	4.5	10.6	8.7	12.1	107
		Urine	1–150	0.989	7.119	0.054	0.8	1	17.4	11.3	15	14.6	9	15.6	9.4	10.1	8	114.6
11OH-9(R)HHC	Δ^9^-THCd_3_	Blood	1–30	0.999	0.063	0.826	0.8	1	9.1	5	7.1	7.4	3.4	2.1	11.4	3.2	11.2	97.4
		OF	1–30	1.000	2.921	0.230	0.8	1	10.6	9.1	3.2	8.8	3	7.6	9.5	3.4	9.7	98.8
		Urine	1–150	0.997	5.752	0.108	0.8	1	14.2	8.8	6.5	12.4	6	4.5	7.6	6.4	14.6	102.5
11OH-9(S)HHC	Δ^9^-THCd_3_	Blood	1–30	0.999	0.028	0.875	0.8	1	6.4	9.8	9.9	10.7	4.5	3.5	9.9	5.9	7.9	81.7
		OF	1–30	0.999	2.91	0.230	0.5	1	11.1	9.6	10.1	10	5.1	5.3	15.1	6.4	14	102
		Urine	1–120	0.997	5.7510	0.1115	0.8	1	12.1	7.0	9.4	14.3	12.7	6.5	12.3	5.1	10.9	114.3
11nor-9(R)COOH HHC	THC COOHd_3_	Blood	1–30	0.9915	4.518	0.124	0.5	1	10.5	6.4	11.3	9.9	2.6	8.5	14.2	8.5	7.5	96.9
		OF	1–30	0.998	3.03	0.224	0.2	1	4.3	4.4	12.1	7	2.6	10.1	7.5	3.4	5	107
		Urine	1–120	0.999	4.667	0.101	0.5	1	16.7	8.3	15.5	14.1	10	11.1	14.3	5.2	11.5	111.1
11nor-9(S)COOH HHC	THC COOHd_3_	Blood	1–50	0.996	3.232	0.214	0.5	1	9.9	7.2	15.1	8.9	2.5	6.9	13	4.7	6.7	89.9
		OF	1–30	0.999	4.406	0.171	0.2	1	6.6	4.5	12.5	7	3	9.8	13.1	4.5	9.9	97
		Urine	1–120	0.999	5.4430	0.101	0.5	1	17.1	2.1	4.3	12.6	6	14.2	17.6	6.1	10.5	102.6

Abbreviations: Δ^9^-THCd_3_, Δ^9^-tetrahydrocannabinol-d_3_; 11nor-9(R)COOH HHC, 11nor-carboxy-9(R)-HHC; 11nor-9(S)COOH HHC, 11nor-carboxy-9(S)-HHC; CV, coefficient of variation; HHC, hexahydrocannabinol; hQC, high quality control; LOD, limit of detection; LOQ, Limit of Quantification; lQC, low quality control; mQC, medium quality control; OF, oral fluid; and r^2^, determination coefficient.

**Table 2 molecules-29-03440-t002:** Analytical results of 10 real samples of whole blood, oral fluid, or urine from HHC consumers.

Matrix	Blood (ng mL^−1^)				OF (ng mL^−1^)			Urine (ng mL^−1^)		
Sample	P1	P5	P6	P7	P2	P3	P4	P9	P8	P10
9(R)-HHC	1.4	n.d.	1.6	2.5	68.8	68.1	295.6	2.7	3.6	7.4
9(S)-HHC	n.d.	n.d.	n.q.	1.6	27.8	27.3	69.7	3.3	2.1	3.0
9αOH-HHC	n.d.	n.d.	n.q.	1.7	n.d	n.d.	n.d.	n.d.	n.d.	n.d.
9βOH-HHC	1.9	n.d.	n.d.	4.4	n.d.	n.d.	n.d.	4.3	n.d.	12.4
8(R)OH-9(R)HHC	5.5	n.d.	6.3	10.9	n.d.	n.d.	n.d.	18.2	12.7	4.3.
8(S)OH-9(S)HHC	n.d.	n.d.	n.d.	n.d.	n.d.	n.d.	n.d.	n.d.	3.4	n.d.
11OH-9(R)HHC	1.3	1.4	n.q.	1.6	n.d.	n.d.	n.d.	62.2	n.d.	29.5
11OH-9(S)HHC	n.d.	n.d.	n.d.	n.d.	n.d.	n.d.	n.d.	7.9	n.d	1.4
11nor COOH-9(S)HHC	n.d.	n.d.	1.5	n.d.	n.d.	n.d.	n.d.	n.d.	n.d.	n.d.
11nor COOH-9(R)HHC	9.7	14.0	13.6	8.6	n.d.	n.d.	n.d.	6.0	6.8	3.4

Abbreviations: HHC, hexahydrocannabinol; 11nor-9(R)COOH HHC, 11nor-carboxy-9(R)-HHC; 11nor-9(S)COOH HHC, 11nor-carboxy-9(S)-HHC; OF, oral fluid; n.d., not detected (<Limit of detection); and n.q., not quantifiable (<Limit of Quantification).

**Table 3 molecules-29-03440-t003:** Retention time, qualitative and quantitative multiple reaction monitoring transitions, and relative collision energy of the target analytes and the internal standards.

Analyte	Rt	MRM Transitions			
		Quantitative	CE	Qualitative	CE
9(R)-HHC	9.263	388 > 345	10	388 > 332	10
9(S)-HHC	9.614	388 > 345	10	388 > 332	10
9αOH-HHC	10.081	404 > 371	20	404 > 386	10
8(R)OH-9(R)HHC	10.600	476 > 371	10	476 > 461	15
9βOH-HHC	10.643	476 > 461	15	146 > 130	10
8(S)OH-9(S)HHC	10.673	476 > 371	10	476 > 393	10
11OH-9(R)HHC	11.282	476 > 371	10	476 > 461	15
11OH-9(S)HHC	11.131	476 > 461	15	476 > 108	10
11nor-9(R)COOH HHC	11.621	490 > 434	10	490 > 385	5
11nor-9(S)COOH HHC	11.537	490 > 434	10	490 > 385	5
THC COOHd_3_	11.882	374 > 292	10	-	
Δ^9^-THCd_3_	9.544	389 > 371	10	-	-

Abbreviations: Δ^9^-THCd3, tetrahydrocannabinol-d3; CE, collision energy; HHC, hexahydrocannabinol; MRM, multiple reaction monitoring; and Rt, retention time.

## Data Availability

Data is unavailable due to privacy or ethical restriction.

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
