# Peer review of "QuEChERS Extraction and Simultaneous Quantification in GC-MS/MS of Hexahydrocannabinol Epimers and Their Metabolites in Whole Blood, Urine, and Oral Fluid"

_molecules, 2024, doi:10.3390/molecules29143440_

Round 1

Reviewer 1 Report

Comments and Suggestions for Authors

Dear Authors,

Manuscript titled 'QuEChERS extraction and simultaneous quantification in GC-2 MS/MS of hexahydrocannabinol epimers and their metabolites 3 in whole blood, urine, and oral fluid' describes the necessary and validated method for determining the increasingly used psychoactive HHC and its metabolites in various blood, urine and oral fluid matrices.

Although the analysis of 10 real samples may not be sufficient for a reliable assessment of metabolic pathways observed in various biological matrices (blood, urine, oral fluid), adding a drawing with possible transformations of HHC to metabolites would be a valuable addition to the manuscript.

In the work, I found minor errors in the names of HHC epimers and the lack of spaces when citing the literature (document with corrections attached).

Kind regards.

Author Response

Comment 1: Although the analysis of 10 real samples may not be sufficient for a reliable assessment of metabolic pathways observed in various biological matrices (blood, urine, oral fluid), adding a drawing with possible transformations of HHC to metabolites would be a valuable addition to the manuscript.

Response 1: In agreement with the reviewer comment, figure 4 (page 14) was added, showing the possible separate metabolic fate of each HHC epimers.

Comment 2: In the work, I found minor errors in the names of HHC epimers and the lack of spaces when citing the literature (document with corrections attached).

Response 2: We would like to thank the reviewer for pointing out all the typos which were all corrected accordingly, and reported in bold in the revised manuscript.

Reviewer 2 Report

Comments and Suggestions for Authors

In this manuscript, the authors evaluated concentrations of HHC and its metabolites in blood, oral fluid, and urine of case samples. This is useful information for forensic toxicologists. However, the reviewer has some questions. When the questions are resolved, the manuscript should be accepted.

1.       In oral fluid, only unchanged HHC isomers were detected. The reviewer thought that it simply attached the smoked e-liquid. The reviewer think that oral fluid does not become true evidence of HHC consumption. Please refute it.

2.       Were the glucuronides of the carboxylated metabolites of HHC (HHC-COOH) truly hydrolyzed by the enzyme? At least, the authors should indicate that the hydrolytic condition the authors used can hydrolyze glucuronide of delta9-THC-COOH.

3.       Why did not the authors select conventional liquid-liquid extraction under acidic condition? Acetonitrile is not always eco-friendly.

4.       Please describe TMS derivatization reagent. At least, the reagent the author used was not simple BSFTA but BSTFA with trimethylchlorosilane.

5.       In relationship with question 4, the reviewer thought that the reason of not proceeding of trimethylsilylation of 9alpha-OH-HHC was inappropriate TMS reagent. Did the authors examined different concentration of trimethylchlorosilane?

Author Response

Comment1: In oral fluid, only unchanged HHC isomers were detected. The reviewer thought that it simply attached the smoked e-liquid. The reviewer think that oral fluid does not become true evidence of HHC consumption. Please refute it.

Response 1: We would like to thank the reviewer for the interesting question. Several studies have already been published in the literature about the excretion of the sole parent drugs into the oral fluid from the hematic stream, depending on the physicochemical properties of the drug and the characteristics of OF and blood. Furthermore, a recent study (Di Trana et al., 2024) on HHC epimers disposition in different biological matrices shows the time course of HHC epimers excretion in this alternative matrix, confirming that the parent drug detected in OF is not an external contamination from the smoked substance. In this context, we provided a brief explanation of the mechanism of drug disposition in oral fluid on page 9, lines 274-277.

Comment 2: Were the glucuronides of the carboxylated metabolites of HHC (HHC-COOH) truly hydrolyzed by the enzyme? At least, the authors should indicate that the hydrolytic condition the authors used can hydrolyze glucuronide of delta9-THC-COOH.

Response 2: We would like to thank the reviewer for the comment. The glucuronides hydrolysis step was developed from the routine extraction protocol for cannabinoids detection in whole blood and urine, currently in use in our laboratory. In agreement with the reviewer, we provided a brief explanation on page 6, lines 158-160.

Comment 3: Why did not the authors select conventional liquid-liquid extraction under acidic condition? Acetonitrile is not always eco-friendly.

Response 3: We would like to thank the reviewer for the comment. According to what reported by Kobidze et al. (2024), a comprehensive liquid/liquid extraction to extract all the considered metabolites should include two different steps at acidic and basic conditions. Otherwise, the modified QuEChERS protocol proved to be efficient in only one step. Furthermore, liquid/liquid extraction is usually performed with toxic organic solvents, such as chloroform, dichloromethane or hexane:ethyl acetate mixtures, which are more pollutant than acetonitrile. Moreover, QuEChERS allowed the extraction of all analytes (parent compound and metabolites) without any pH adjustment. Thus, QuEChERS extraction with acetonitrile demonstrated to be the best compromise. A brief explanation was added at page 9, lines 263-265

Comment 4: Please describe TMS derivatization reagent. At least, the reagent the author used was not simple BSFTA but BSTFA with trimethylchlorosilane.

Response 4: In agreement with the reviewer comment, we specified the precise composition of the derivatization agent as purchased from Sigma-Aldrich on page 5, lines 126- 127.

Comment 5: In relationship with question 4, the reviewer thought that the reason of not proceeding of trimethylsilylation of 9alpha-OH-HHC was inappropriate TMS reagent. Did the authors examined different concentration of trimethylchlorosilane?

Response 5: We would like to thank the reviewer for the comment. Since the derivatization agent was purchased as a mixture of N,O-Bis(trimethylsilyl)trifluoroacetamide with trimethylchlorosilane (99:1, v/v), the only composition studied was the one that we already reported in the manuscript. In agreement to the reviewe, we added a comment on the possible influence of TMS percentage in the derivative mixture for the complete derivatization of the analyte, at lines 240-241, page 9

Round 2

Reviewer 2 Report

Comments and Suggestions for Authors

The authors well revised the manuscript. However, one part (Line 239 TMS) should be corrected to "trimethylchlorosilane". The reviewer thought that the authors did not understand why trimethylchlorosilane was added. The authors should re-study TMS derivatization.

Author Response

Comment 1: The authors well revised the manuscript. However, one part (Line 239 TMS) should be corrected to "trimethylchlorosilane". The reviewer thought that the authors did not understand why trimethylchlorosilane was added. The authors should re-study TMS derivatization.

Reponse 1: We thank the reviewer for pointing this out. In agreement with the reviewer's comment, we rephrased the sentence ( lines 239-241) for better understanding, amending the abbreviation "TMS" to "trimethylchlorosilane" (line 239), as suggested. 
